# Indoor Carrier Phase Positioning Technology Based on OFDM System

**DOI:** 10.3390/s21206731

**Published:** 2021-10-11

**Authors:** Zhenyu Zhang, Shaoli Kang, Xiang Zhang

**Affiliations:** 1School of Electronic and Information Engineering, Beihang University, Beijing 100191, China; 2State Key Laboratory of Wireless Mobile Communications, China Academy of Telecommunications Technology, Beijing 100083, China; kangshaoli@catt.cn (S.K.); zhangxiang4@datangmobile.cn (X.Z.)

**Keywords:** extended Kalman filter, localization, time of arrival, carrier phase, ambiguity resolution

## Abstract

Carrier phase measurement is a ranging technique that uses the receiver to determine the phase difference between the received signal and the transmitted signal. Carrier phase ranging has a high resolution; thus, it is an important research direction for high precision positioning. It is widely used in global navigation satellite systems (GNSS) systems but is not yet commonly used inwireless orthogonal frequency division multiplex (OFDM) systems. Applying carrier phase technology to OFDM systems can significantly improve positioning accuracy. Like GNSS carrier phase positioning, using the OFDM carrier phase for positioning has the following two problems. First, multipath and non-line-of-sight (NLOS) propagation have severe effects on carrier phase measurements. Secondly, ambiguity resolution is also a primary issue in the carrier phase positioning. This paper presents a ranging scheme based on the carrier phase in a multipath environment. Moreover, an algorithm based on the extended Kalman filter (EKF) is developed for fast integer ambiguity resolution and NLOS error mitigation. The simulation results show that the EKF algorithm proposed in this paper solves the integer ambiguity quickly. Further, the high-resolution carrier phase measurements combined with the accurately estimated integer ambiguity lead to less than 30-centimeter positioning error for 90% of the terminals. In conclusion, the presented methods gain excellent performance, even when NLOS error occur.

## 1. Introduction

With the rapid development of industries such as the Internet of Things and industrial control, high-precision indoor positioning technology has become an important issue to be solved. It is challenging to receive valid satellite navigation signals in the indoor environment, and other high-precision positioning technologies need to be studied. In recent years, positioning services based on wireless communications are rapidly developing. The mobile cellular network covers a wide area and is one option for dense urban areas and indoor positioning. Benefit from the advance of 5G technology, high-precision positioning using the wireless access network has become a hot research direction. In wireless networks, traditional ranging-based positioning methods include angle of arrival (AOA), received signal strength (RSS), and time-of-arrival (TOA). Among them, AOA determines location of the user by measuring the angle between the terminal and the base station (BS) [1]. Since measuring angle often requires a sufficient number of antennas at the receiver, the application range of AOA technology is limited. RSS technology needs to establish an accurate signal energy propagation model, making it challenging to achieve high measurement accuracy [2]. TOA-based positioning technology converts arrival time to a distance and then uses the distance information for positioning. TOA has been widely used due to its low requirements of positioning equipment [3]. This paper mainly studies high-precision TOA measurement and positioning algorithms based on orthogonal frequency division multiplex (OFDM) systems.

TOA estimation can be considered as a channel estimation problem. Many documents carry out channel impulse response (CIR) estimation from the time domain or frequency domain perspective in the OFDM multi-carrier system [4,5,6,7,8,9]. Typical schemes are cross-correlation algorithms based on the pseudo-randomness of the transmission sequence, including the maximum criterion algorithm and the threshold algorithm [4]. However, this positioning method is limited by the signal bandwidth and receiver resolution, and it is challenging to achieve sub-meter accuracy. Besides, TOA-based parameter estimation techniques have been widely studied, including multiple signal classification algorithm (MUSIC) [10], Signal Parameters via Rotational Invariant Techniques (ESPRIT) [11], and Space-Alternating Generalized Expectation maximization (SAGE) [12] algorithms. These algorithms are not limited by the system sampling rate but are determined by the search interval for time delay estimation. A smaller search interval will effectively improve the accuracy of TOA estimation but will significantly increase the computational complexity. The vast computational overhead makes it difficult to apply such algorithms to real-time user localization scenarios. Other studies have attempted to apply phase ranging techniques to indoor localization [13,14,15]. However, specific phase emission and measurement devices are challenging to reduce the cost of localization effectively. Phase measurement techniques based on OFDM systems can provide high accuracy positioning measurements while satisfying communication requirements, and therefore are a research direction for phase-based positioning. References [16,17,18] describe methods for distance measurement through the phase difference of subcarriers in OFDM systems. Still, such schemes are often only suitable for LOS propagation or multipath propagation when the Rice factor is high. Carrier phase localization techniques based on wireless cellular networks have been proposed in the literatures [19,20]. Furthermore, carrier phase measurement in multipath environments and the suppression of non-line-of-sight (NLOS) error require further research.

There are two modes for receivers in global navigation satellite systems (GNSS): pseudorandom code (C/A code or P-code) and the carrier phase [21,22]. The ranging principle of pseudorandom code mode is similar to TOA, and the measurement error of pseudorandom code mode is vast. The carrier phase ranging method uses the carrier phase of the measurement signal to extract the propagation distance information. Under line-of-sight (LOS) conditions, the carrier phase’s measurement error is a small fraction of the carrier wavelength and can reach the centimeter range. However, the carrier phase measurement includes unknown integer ambiguity: the distance between the user and the BS in terms of carrier wavelength can be divided into an integral part of the wavelength plus a fractional function. During the initialization of positioning, the phase measurement is in the range [0,2π], so only fractional multiples of the distance can be measured, which causes the problem of integer ambiguity. Once this problem is solved, the carrier phase positioning can meet the requirements of high accuracy.

Inspired by reference [19], the positioning accuracy might be significantly improved if the carrier phase technology can be extended in the indoor location system. Compared with GNSS positioning, wireless networks can work in challenging scenarios and have more flexible carrier frequency configurations, fewer error sources, and more minor path losses. These characteristics constitute the advantages of supporting carrier phase technology in wireless networks. However, despite these advantages, there are many challenges while applying carrier phase positioning in wireless networks: Multipath and NLOS propagation in the indoor environment, fast resolution of integer ambiguities in wireless networks, etc.

In summary, this paper proposes a carrier phase-ranging scheme based on the OFDM system. Under the premise of high accuracy ranging, this paper focuses on two aspects: carrier phase measurement in a multipath environment and how to solve the integer ambiguity quickly and accurately. First, we model the carrier phase measurement in a multipath environment and analyze the integer ambiguity generation. Second, we propose an extended Kalman filter (EKF) for solving the integer ambiguity. The EKF-based algorithm can solve the position of the terminal while solving the integer ambiguity. Further, we describe how to utilize the EKF to mitigate the errors caused by NLOS propagation.

The overall structure of this study contains five chapters. Section 2 describes the model for studying TOA and phase estimation in OFDM systems. In Section 3, we propose an EKF that combines carrier phase and TOA measurements to enhance positioning accuracy and reduce the impact of NLOS error on mobile positioning. Numerical simulation results are presented in Section 4 to prove the effectiveness of the new methods. In Section 5, we summarize conclusions drawn from this paper.

In this paper, vectors and matrices are denoted by boldface lower-case letters and boldface upper-case letters. The superscript ${\left[.\right]}^{T}$ denotes the transpose. The superscript ${\left[.\right]}^{ij}$ and the subscript ${\left[.\right]}_{ab}$ represent, respectively, the single-difference (SD) between the transmitters and between the receivers.

## 2. Ranging System

Consider OFDM transmission with *N* subcarriers, subcarrier spacing $\Delta f_{SCS}$ and sampling interval $T_{S}=1/\left(N\Delta f_{SCS}\right)$. OFDM transmission is block oriented. Assume *N* quadrature-amplitude modulation (QAM) symbols $X_{k}^{m},k\in \left\{1,\ldots ,N\right\}$ are grouped into a vector $X^{m}={\left[X_{1}^{m},\ldots ,X_{N}^{m}\right]}^{T}$ and transmitted in the *m*-th OFDM symbol in a slot. A unitary inverse discrete-time Fourier transform (IDFT) on $X^{m}$ gives a continuous time representation of the complex envelope of an OFDM symbol of duration $T=NT_{s}=1/\Delta f_{SCS}$ (note: here *T* does not include cyclic prefix).
(1)$$x^{m}\left(t\right)=\frac{1}{\sqrt{N}}\sum_{k=1}^{N}X_{k}^{m}e^{j2\pi (k-1)t/T};0\leq t\leq T,$$
the time-domain signal $x^{m}\left(t\right)$ is up-converted to the carrier frequency $f_{c}$ for transmission.
(2)$$\begin{array}{cc} s^{m}\left(t\right) & =x^{m}\left(t\right)e^{j2\pi f_{c}t}\\ & =\frac{1}{\sqrt{N}}\sum_{k=1}^{N}X_{k}^{m}e^{\left.j2\pi (\left(k-1\right)/T+f_{c}\right)t};\;0\leq t\leq T. \end{array}$$

Assume the channel is the quasi-static channel, i.e., the channel does not change during the transmission of one OFDM symbol, the quasi-static channel can then be described by a time discrete CIR $h={\left[h_{0}\left(t\right),h_{1}\left(t\right),\ldots ,h_{L_{p}-1}\left(t\right)\right]}^{T}$, multipath channel model can be expressed as:(3)$$h\left(t,\tau \right)=\sum_{l=1}^{L_{p}}h_{l}\left(t\right)\delta \left(t-\tau_{l}\left(t\right)\right)+h_{d}\left(t,\tau \right),$$
$L_{p}$ is the total number of paths which include one LOS path and $L_{p}-1$ NOLS paths, $h_{l}\left(t\right)$ is the gain for the *l*-th path, $\delta \left(t-\tau_{l}\left(t\right)\right)$ is the Dirac delta function, $\tau_{l}\left(t\right)$ is the TOA of the *l*-th path, $h_{d}\left(t,\tau \right)$ are the diffuse multipath components (DMC) [23], which represent the non-discrete part of the channel. The received signal after passing through the multipath channel can be expressed as:(4)$$y^{m}\left(t\right)=s^{m}\left(t\right)⊗h\left(t,\tau \right)+w_{n}^{m}=\int_{-\infty }^{\infty }s^{m}\left(\xi \right)h\left(t-\xi ,\tau \right)d\xi +w^{m},$$
$w^{m}$ is the color noise consisting of $w_{n}^{m}$ and $s^{m}\left(t\right)⊗h_{d}\left(t,\tau \right)$, where $w_{n}^{m}∼N\left(0,\sigma^{2}\right)$ is the complex additive noise with zero mean and $\sigma^{2}$ variance. If the received signal contains color noise, it is necessary to consider the use of whitening filters to convert the color noise to white noise [24]. Furthermore, the received *m*-th OFDM symbol can be expressed by:(5)$$\begin{array}{cc} y^{m}\left(t\right) & =\frac{1}{\sqrt{N}}\sum_{k=1}^{N}X_{k}^{m}\left[\sum_{l=1}^{L_{p}}h_{l}\left(t\right)e^{-j2\pi \left(f_{c}+\frac{k-1}{T}\right)\tau_{l}\left(t\right)}\right]e^{j2\pi \left(f_{c}+\frac{k-1}{T}\right)t}+w^{m}\\ & =\frac{1}{\sqrt{N}}\sum_{k=1}^{N}X_{k}^{m}\left[\sum_{l=1}^{L_{p}}h_{l}\left(t\right)e^{-j2\pi \left(f_{c}+\frac{k-1}{T}\right)\tau_{l}\left(t\right)}+w_{k}^{m}\right]e^{j2\pi \left(f_{c}+\frac{k-1}{T}\right)t} \end{array}.$$

After down-conversion and removal of the samples of the received signal which belong to the cyclic prefix, the received signal $y^{m}\left(t\right)$ is converted into a discrete time domain signal $y^{m}\left[n\right]$:(6)$$\begin{array}{c} y^{m}\left[n\right]=\frac{1}{\sqrt{N}}\sum_{k=1}^{N}X_{k}^{m}\left[\sum_{l=1}^{L_{p}}h_{l}\left[nT_{s}\right]e^{-j2\pi \left(f_{c}+\frac{k-1}{T}\right)\tau_{l}\left[nT_{s}\right]}+w_{k}^{m}\right]e^{j2\pi \frac{n(k-1)}{N}} \end{array}.$$

### 2.1. Conventional Cross-Correlation TOA Estimator

Signal arrival time needs to be obtained from reference signals. In Long Term Evolution (LTE) Release 9, positioning reference signals (PRS) were used to improve TOA-based positioning. PRS are pseudo-random sequences with good autocorrelation. With the help of the autocorrelation characteristics of the PRS sequence, it is easier to find the direct path in the environment of multipath transmission. The cross-correlation expression is:(7)$$\begin{array}{cc} R_{xy} & =\sum_{n=1}^{N}\bar{x^{m}\left[n-\tau \right]}y^{m}\left[n\right]\\ \; & =h_{1}\left[nT_{s}\right]e^{-j2\pi f_{c}\tau_{1}\left[nT_{s}\right]}R_{m}^{xx}\left[\tau -\tau_{1}\left[nT_{s}\right]\right]+\sum_{i=2}^{L}h_{i}\left[nT_{s}\right]e^{-j2\pi f_{c}\tau_{i}\left[nT_{s}\right]}R_{m}^{xx}\left[\tau -\tau_{i}\left[nT_{s}\right]\right]\\ & +R_{m}^{xw}\left[\tau \right],\tau \in \left[1,...N\right]; \end{array}$$
where $\bar{\left(.\right)}$ denotes the complex conjugate function, x[n] is the replica of the transmitted PRS. Furthermore, based on the autocorrelation of the PRS series, we have:(8)$$\begin{array}{cc} R_{m}^{xx}\left[\tau \right] & =\sum_{n=1}^{N}\bar{x^{m}\left[n-\tau \right]}x^{m}\left[n\right]=\delta \left[\tau \right]\\ R_{m}^{xw}\left[\tau \right] & =\sum_{n=1}^{N}\bar{x^{m}\left[n-\tau \right]}w^{m}\left[n\right]\approx 0 \end{array}.$$
where $w^{m}\left[n\right]$ is the downsampled additive noise. From Equation (Equation 7), it can be seen that the magnitude of the correlation function is affected by the carrier phase $2\pi f_{c}\tau_{i}\left[nT_{s}\right]$. To exclude this effect, we use $|R_{xy}\left[\tau \right]|$ instead of $R_{xy}\left[\tau \right]$ for TOA estimation. Taking the threshold method as an example, TOA is determined by estimating the time delay of the first (earliest) peak in the magnitude of the normalized cross-correlation function above a certain threshold [4].
(9)$$\hat{\tau }=\underset{\tau }{\arg\min}\left\{\frac{|R_{xy}\left[\tau \right]|}{\max\left\{\left|R_{xy}\right|\right\}}\geq \zeta \right\},$$
here, ζ is the preset threshold. Correlation profile-based methods can estimate the propagation delay of the first path in a multipath environment. Still, due to the limited sampling rate of the system, the measurement accuracy of this method is low. Rewriting $\hat{\tau }$ to $T_{r}^{i}$ and introducing terminal *r* and the BS *i*, then the estimated TOA can be modeled as [25]:(10)$$T_{r}^{i}=\left(d_{r}^{i}+w_{r,T}^{i}\right)/c.$$

$T_{r}^{i}$ (known) is the TOA measurement from terminal *r* to BS *i* (unit:s).*c* is the speed of radio waves in vacuum, 299,792,458 (unit: m/s).$d_{r}^{i}=\sqrt{{\left(x_{i}-x_{}\right)}^{2}+{\left(y_{i}-y_{}\right)}^{2}}$ (unknown) is the geometric distance between the antennas of transmitter *i* and receiver *r* (unit: m).$\left(x_{i},y_{i}\right)$ (known) is the two-dimensional vector giving the coordinates of BS *i*.(x,y) (unknown) is the location of the terminal to be solved.$w_{r,T}^{i}∼N\left(0,\sigma_{r,i}^{2}\right)$ (unknown) is a random Gaussian variable accounting for the residual estimation error (unit: m).

### 2.2. High-Precision TOA Estimation Scheme Based on Carrier Phase

From Equation (Equation 5), combine the known PRS signal, the frequency domain channel response is written as:(11)$$H^{m}\left(k\right)=\sum_{l=1}^{L_{p}}h_{l}\left[nT_{s}\right]e^{-j2\pi \frac{k-1}{N}{\dot{\tau }}_{l}-j\phi_{l}}+w_{k}^{m},$$
where ${\dot{\tau }}_{l}=N\Delta f_{SCS}\tau_{l}\left[nT_{s}\right]$ is the transmission delay in units of sampling interval. $\phi_{l}=2\pi f_{c}\tau_{l}\left[nT_{s}\right]$ is the phase shift caused by free-space propagation.

As can be seen from Equation (Equation 11), the distance between the BS and the terminal is reflected in each subcarrier phase. However, due to the signal aliasing of multiple transmission paths, it is difficult to directly estimate phase information of the first path from the unprocessed subcarrier phase. Therefore, we convert the frequency domain channel response to the time domain for further analysis. Furthermore, when the distance (in units of sampling interval) is not an integer multiple of the sampling interval, the time domain channel response is subject to energy leakage [26]:(12)$$h_{n}^{m}=\frac{\sin\left(\pi {\dot{\tau }}_{l}\right)}{\sqrt{N}\sin\left(\frac{\pi }{N}\left({\dot{\tau }}_{l}-n\right)\right)}\sum_{L_{p}}h_{l}\left[nT_{s}\right]e^{-j\frac{\pi }{N}\left(n+\left(N-1\right){\dot{\tau }}_{l}\right)-j\phi_{l}}.$$

Much of the literature [27,28] describes using Equation (Equation 7) or (Equation 9) to find the integer multiple sampling points closest to the transmission delay, denoted as $\left[{\dot{\tau }}_{1}\right]$, and [.] is a rounding function. The time domain signal is processed to eliminate the effects of multipath effects. The window can be expressed as: for $n\in [\left[{\dot{\tau }}_{1}\right]-\frac{W}{2},\left[{\dot{\tau }}_{1}\right]+\frac{W}{2}],$${\tilde{h}}_{n}^{m}=h_{n}^{m},$ else ${\tilde{h}}_{n}^{m}=0.$ Furthermore, *W* is the length of the window.

We use the tapped delay line model to characterize the frequency-selective channel, and each tap represents a different channel delay in units of the sampling interval of the receiver. Figure 1 shows a schematic of the window at W=0. Based on the correlation of the transmit sequence, the terminal can determine the arrival delay of the direct path. The manipulation of the power delay profile further eliminates the effects of multipath. Additionally, it is worth noting that actual distance to the BS in this example is 12.4 sampling intervals. Due to the limitation of the system sampling rate, the TOA obtained by the cross-correlation algorithm is 12 sampling intervals, which generates a significant measurement error. Furthermore, as shown in the figure, the distance is a non-integer multiple of the sampling interval resulting in leakage between taps.

For the convenience of analysis, we consider W=0 and the window can be expressed as:(13)$$\left\{\begin{array}{c} {\tilde{h}}_{n}^{m}=\frac{h_{1}\left[nT_{s}\right]\sin\left(\pi {\dot{\tau }}_{1}\right)}{\sqrt{N}\sin\left(\frac{\pi }{N}\left(\dot{\tau_{1}}-n\right)\right)}e^{-j\frac{\pi }{N}\left(n+\left(N-1\right){\dot{\tau }}_{1}\right)-j\phi_{1}},n=\left[{\dot{\tau }}_{1}\right]\\ {\tilde{h}}_{n}^{m}=0,n\in N_{|{\dot{\tau }}_{1}} \end{array}\right.,$$
here, $N_{|{\dot{\tau }}_{1}}=\left\{1,2,\left[{\dot{\tau }}_{1}\right]-1,\left[{\dot{\tau }}_{1}\right]+1,\cdots ,N\right\}.$ The frequency domain channel response after the window function can be written as:(14)$$H_{k}^{m}=\frac{h_{1}\left[nT_{s}\right]\sin\left(\pi {\dot{\tau }}_{1}\right)e^{-j\frac{\pi }{N}\left(\left[{\dot{\tau }}_{1}\right]+\left(N-1\right){\dot{\tau }}_{1}\right)-j\phi_{1}-j2\pi \frac{\left[{\dot{\tau }}_{1}\right]}{N}k}}{\sqrt{N}\sin\left(\frac{\pi }{N}\left({\dot{\tau }}_{1}-\left[{\dot{\tau }}_{1}\right]\right)\right)}.$$

The time-domain window eliminates the effect of multipath on the carrier phase, but introduces additional problems at the same time.

(1) The phase difference between sub-carriers $e^{-j2\pi \frac{\left[{\dot{\tau }}_{1}\right]}{N}}$ can no longer accurately reflect the distance. The system resolution also limits the time delay measured through the subcarrier phase due to the effect of the time domain window.

(2) The time-domain window processing introduces some phase noise. For example, $e^{-j\frac{\pi }{N}\left(\left[{\dot{\tau }}_{1}\right]+\left(N-1\right){\dot{\tau }}_{1}\right)}$. Furthermore, the sign of $\frac{\sin\left(\pi {\dot{\tau }}_{1}\right)}{\sin\left(\frac{\pi }{N}\left(\dot{\tau_{1}}-\left[{\dot{\tau }}_{1}\right]\right)\right)}$ will also affect the phase of sub-carriers.

It can be proved that the subcarrier phase of k=N/2 can effectively reflect the distance information from the terminal to the BS. Furthermore, the channel frequency response of subcarrier k=N/2 can be approximated as:(15)$$H_{k=N/2}^{m}=\frac{h_{1}\left[nT_{s}\right]\sin\left(\pi {\dot{\tau }}_{1}\right)}{\sqrt{N}\sin\left(\frac{\pi }{N}\left({\dot{\tau }}_{1}-n\right)\right)}e^{-j\pi {\dot{\tau }}_{1}-j\phi_{1}}+w_{N/2}^{m},$$
the proof is given in the Appendix A. Therefore the phase at k=N/2 can be written as:(16)$$\hat{\phi }=-angle\left(e^{-j\pi \dot{\tau }-j\phi_{1}-{\hat{w}}_{P}}\right);0\leq \hat{\phi }\leq 2\pi ,$$
here, ${\hat{w}}_{P}$ is the phase noise caused by $w_{N/2}^{m}$. Due to the trigonometric function properties, the part beyond 2π cannot be found when solving for the phase; thus, the integer ambiguity arises. Considering the phase shifts experienced in the channel, e.g., phase noise, base on ${\dot{\tau }}_{1}=N\Delta f_{SCS}\tau_{1}\left[nT_{s}\right]$ and $\phi_{1}=2\pi f_{c}\tau_{1}\left[nT_{s}\right]$, the phase-based ranging can be written as:(17)$$\hat{\phi }+2\pi N_{I}=\pi N\Delta f_{SCS}\tau_{1}\left[nT_{s}\right]+2\pi f_{c}\tau_{1}\left[nT_{s}\right]+{\hat{w}}_{P},$$
here, $N_{I}$ is the unknown integer ambiguity. Divide Equation (Equation 17) by 2π and simplify the equation:(18)$$\begin{array}{cc} \phi & =\frac{N}{2}\Delta f_{SCS}\tau_{1}\left[nT_{s}\right]+f_{c}\tau_{1}\left[nT_{s}\right]-N_{I}+w_{P}\\ \; & =\frac{d}{\lambda }-N_{I}+w_{P}, \end{array}$$
here, $\phi =\frac{\hat{\phi }}{2\pi }$ is the normalized phase measurement, $w_{P}=\frac{{\hat{w}}_{P}}{2\pi }$ is the normalized phase noise, $d=c\tau_{1}\left[nT_{s}\right]$ is the geometric distance between the antennas of transmitter and receiver, $\lambda =\frac{c}{f_{c}+\frac{N}{2}\Delta f_{SCS}}$ is the equivalent wavelength. Further, introducing terminal *r* and the BS *i*, we have:(19)$$\phi_{r}^{i}=\frac{d_{r}^{i}}{\lambda }-N_{r}^{i}+w_{r,P}^{i}.$$

$\phi_{r}^{i}$ (known) is the carrier phase measurement (unit: carrier circle).λ (known) is the wavelength calculated from *c*, $f_{c}$, *N*, and $\Delta f_{SCS}$, (unit: m).$N_{r}^{i}$ (unknown) is the integer ambiguity (unit: carrier circle).$w_{r,P}^{i}∼N\left(0,{\tilde{\sigma }}_{r,i}^{2}\right)$ (unknown) is the residual estimation error (unit: carrier circle).

The time delay from the user to the BS can be deduced by measuring the phase of the N/2 subcarrier. Equation (Equation 16) shows that the system sampling rate does not limit the carrier phase measurement, and thus the accuracy of the carrier phase-based ranging technique is high. Furthermore, we use the phase-lock-loop (PLL) [29] to measure the carrier phase. At the initial locking moment of the PLL, the carrier phase measurement is between [0,1]. After that, the change of user position will be reflected in the measured phase (continuous phase tracking allows carrier phase more than 1 or less than 0), thus ensuring that the integer ambiguity is constant during the user positioning. However, since the integer ambiguity is unknown, the carrier phase measurements are challenging to be used directly for user location solutions. Therefore, we propose a location algorithm combining carrier phase and TOA measurements in the following.

## 3. Positioning Algorithm

The ambiguity resolution is one of the primary problems in carrier phase measurement. In this section, we propose an EFK algorithm based on TOA and carrier phase measurements. This algorithm can estimate the position while estimating the integer ambiguity.

### 3.1. TOA and Carrier Phase Measurements

According to Equations (Equation 10) and (Equation 19), further considering the non-ideal factors such as clock error and NLOS error, the TOA measurements and carrier phase between the *i*-th BS and user equipment (UE) *r* at a specific epoch can be written as:(20)$$\begin{array}{c} T_{r}^{i}=\left(d_{r}^{i}+m_{r}^{i}+w_{r,T}^{i}\right)/c+\delta t^{i}-\delta t_{r}\;\\ \phi_{r}^{i}=\frac{d_{r}^{i}+c\left(\delta t^{i}-\delta t_{r}\right)+m_{r}^{i}}{\lambda }-N_{r}^{i}+w_{r,P}^{i} \end{array},$$

$\delta t^{i}$ (unknown) is the clock error of the transmitter *i* (unit: s).$\delta t_{r}$ (unknown) is the clock error of the receiver *r* (unit: s).$m_{r}^{i}$ (unknown) represents the channel bias introduced by NLOS reflections (unit: m).

The SD of the TOA and carrier phase measurements from the receiver *r* by measuring the signals from two transmitters *i* and *j* can be expressed as:(21)$$\begin{array}{c} T_{r}^{ij}=(d_{r}^{ij}+m_{r}^{ij}+w_{r,T}^{ij})/c+\delta t^{ij}\;\\ \phi_{r}^{ij}=\frac{d_{r}^{ij}+c\delta t^{ij}+m_{r}^{ij}}{\lambda }-N_{r}^{ij}+w_{r,P}^{ij} \end{array},$$
where the double superscript “ij” indicates the differential operation between transmitters *i* and *j*, i.e., $s_{r}^{ij}=s_{r}^{i}-s_{r}^{j};s\in \left\{T,\phi ,d,\delta t,N,m,w\right\}$. According to Equations (Equation 10) and (Equation 19), the measurement noise $w_{r,T}^{ij}$ and $w_{r,P}^{ij}$ are still independent Gaussian noise with following distributions, i.e.,
(22)$$E\left[w_{r,T}^{ij},w_{r,T}^{kj}\right]=\left\{\begin{array}{cc} \sigma_{r,i}^{2}+\sigma_{r,j}^{2}; & i=k\\ \sigma_{r,j}^{2}; & i\neq k \end{array}\right..$$

The SD operation of Equation (Equation 21) removes the measurement errors common to the receiver, e.g., the receiver clock offset $\delta t_{r}$. Furthermore, the double-difference (DD) TOA and carrier phase measurements from two transmitters *i* and *j*, and two receivers *r* and *u* can be expressed as:(23)$$\begin{array}{c} T_{ru}^{ij}=\left(d_{ru}^{ij}+m_{ru}^{ij}+w_{ru,T}^{ij}\right)/c\;\\ \phi_{ru}^{ij}=\frac{d_{ru}^{ij}+m_{ru}^{ij}}{\lambda }-N_{ru}^{ij}+w_{ru,P}^{ij} \end{array},$$
where the double superscript “ij” indicates the differential operation between transmitters *i* and *j*, and double subscript “ru” indicates the differential operation between receivers *r* and *u*, $s_{ru}^{ij}=s_{r}^{ij}-s_{u}^{ij}=\left(s_{r}^{i}-s_{r}^{j}\right)-\left(s_{u}^{i}-s_{u}^{j}\right);s\in \left\{T,\phi ,d,\delta t,N,m,w\right\}$. DD measurement noise $w_{ru,T}^{ij}$ and $w_{ru,P}^{ij}$ are no longer independent Gaussian noise. Assume the transmitter *j* is selected as the reference, we have:(24)$$E\left[w_{ru,T}^{ij},w_{ru,T}^{kj}\right]=\left\{\begin{array}{c} \sigma_{r,j}^{2}+\sigma_{u,j}^{2};i\neq k\\ \sigma_{r,i}^{2}+\sigma_{u,i}^{2}+\sigma_{r,j}^{2}+\sigma_{u,j}^{2};i=k \end{array}\right..$$

DD operation removes the measurement biases related to the transmitters and the receivers, such as the transmitter clock offsets and receivers clock offsets. We introduce the concept of reference device, where it is assumed that *u* is the reference device and that the location of terminal *u* is known. It can be seen by Equation (Equation 23) that the introduction of the reference device helps to eliminate the clock error. We can construct the SD measurements from the DD measurements, which are not impacted by the receiver and the transmitter clock biases. Given that $d_{u}^{ij}$ can be obtained from the known locations of the reference device *u* and the BSs, we can construct the SD measurements ${\hat{T}}_{r}^{ij}$ and $\Phi_{r}^{ij}$:(25)$$\begin{array}{c} {\hat{T}}_{r}^{ij}≜cT_{ru}^{ij}+d_{u}^{ij}=d_{r}^{ij}+m_{ru}^{ij}+w_{ru,T}^{ij}\\ \Phi_{r}^{ij}≜\phi_{ru}^{ij}+\frac{d_{u}^{ij}}{\lambda }=\frac{d_{r}^{ij}+m_{ru}^{ij}}{\lambda }-N_{ru}^{ij}+w_{ru,P}^{ij} \end{array},$$

Equation (Equation 25) shows the ${\hat{T}}_{r}^{ij}$ and $\Phi_{r}^{ij}$ are not impacted by the receiver and the transmitter clock biases. It is worth noting that the reference device can be either a UE with a known exact location or a BS. For some positioning scenarios, the deployment of additional hardware can cause a significant overhead; therefore, 3GPP has agreed on selecting the reference device, i.e., the device with the known location can be a UE and/or a BS (also known as evolved gNB) [30].

### 3.2. Extended Kalman Filter

For an EKF design, one needs first to define the unknown EKF states. An EKF for carrier phase positioning may include the following EKF states:UE position. EKF for positioning needs to include the states associated with the unknown UE position. The EKF may use the 2D (or 3D) UE position coordinates directly as the EKF states. For example, in the following discussion of the EKF design, we assume the EKF states include a 2D position.UE velocity. With the consideration of UE mobility, the EKF states may also include the UE velocity. The number of states for UE velocity is generally the same as the number of states for UE position.Integer ambiguities. The premise of using the carrier phase for location is to solve integer ambiguities. According to Equation (Equation 25), it is necessary to solve the DD integer ambiguities while solving the user position.

Let the position of the UE at epoch *k* be s(k). In the absence of other information, assume that the velocity of UE keeps constant, the position at the next epoch can be expressed as s(k+1)=s(k)+vT. Furthermore, in the case of no cycle slip, the ambiguities remain consistent in each epoch. Assume the system states include 2D position, 2D velocity, and the DD integer ambiguities are obtained from *m* cells, and the system can be represented as:(26)$$\left\{\begin{array}{c} x\left(k+1\right)=x\left(k\right)+v_{x}\left(k\right)T\;\\ v_{x}\left(k+1\right)=v_{x}\left(k\right)\;\\ y\left(k+1\right)=y\left(k\right)+v_{y}\left(k\right)T\;\\ v_{y}\left(k+1\right)=v_{y}\left(k\right)\;\\ N_{ru}^{ij}\left(k+1\right)=N_{ru}^{ij}\left(k\right) \end{array}\right.$$

Assume the *j*-th cell is selected as the reference cell. The EKF state vector x can be expressed as follows:(27)$$\begin{array}{c} x≜{\left[s,v,N\right]}^{T}\\ ={\left[x,y,v_{x},v_{y},N_{ru}^{1j},\ldots ,N_{ru}^{(j-1)j},N_{ru}^{(j+1)j},\ldots ,N_{ru}^{mj}\right]}^{T} \end{array},$$
where $s=\left(x,y\right)$ models the UE position; $v=(v_{x},v_{y})$ is the UE velocity, and $N=[N_{ru}^{1j},\ldots ,N_{ru}^{(j-1)j},N_{ru}^{(j+1)j},\ldots ,N_{ru}^{mj}]$ includes the DD integer ambiguities. Based on the selected EKF states, the state transition equation of the discrete EKF for carrier phase positioning can be written as:(28)$$x\left(k+1\right)=F\left(k\right)x\left(k\right)+W_{x}\left(k\right).$$

The one-step state transition matrix is as follows:(29)$$F=\left[\begin{array}{ccc} I(2\times 2) & F_{12} & 0\\ 0 & I(2\times 2) & 0\\ 0 & 0 & I(m-1\times m-1) \end{array}\right],$$
where $F_{12}=\left[\begin{array}{cc} \Delta T & 0\\ 0 & \Delta T \end{array}\right]$, $E[W_{x}]=0$, and $Q=E\left[W_{x}W_{x}^{T}\right]=$$\mathrm{diag}\left(Q_{r};Q_{v};0\left(m-1\times m-1\right)\right)$, $Q_{r}=\mathrm{diag}\left\{\sigma_{x}^{2},\sigma_{y}^{2}\right\}$, $Q_{v}=\mathrm{diag}\left\{\sigma_{v_{x}}^{2},\sigma_{v_{y}}^{2}\right\}$, I represents an identity matrix, and 0 represents a zero matrix. ΔT is the time interval of the state transition of the Kalman filter. $\sigma_{x}^{2},\sigma_{y}^{2}$ and $\sigma_{v_{x}}^{2},\sigma_{v_{y}}^{2}$ represent the uncertainty in the prediction of the UE position and velocity.

The measurement equations of the discrete EKF as:(30)$$Z\left(k+1\right)=h\left(x\left(k+1\right)\right)+W_{Z}\left(k+1\right)$$
(31)$$Z\left(k+1\right)=\left[\begin{array}{c} T\\ \Theta \end{array}\right],T\left(k+1\right)=\left[\begin{array}{c} {\hat{T}}_{r}^{1j}\left(k+1\right)\\ \vdots \\ {\hat{T}}_{r}^{(j-1)j}\left(k+1\right)\\ {\hat{T}}_{r}^{(j+1)j}\left(k+1\right)\\ \vdots \\ {\hat{T}}_{r}^{mj}\left(k+1\right) \end{array}\right]$$
(32)$$\Theta \left(k+1\right)=\left[\begin{array}{c} \Phi_{r}^{1j}\left(k+1\right)\\ \vdots \\ \Phi_{r}^{(j-1)j}\left(k+1\right)\\ \Phi_{r}^{(j+1)j}\left(k+1\right)\\ \vdots \\ \Phi_{r}^{mj}\left(k+1\right) \end{array}\right]$$
(33)$$\begin{array}{c} W_{z}\left(k+1\right)=\left[\begin{array}{c} W_{T}\left(k+1\right)\\ W_{P}\left(k+1\right) \end{array}\right],E\left[W_{z}\right]=0\;\\ R=E\left[W_{z}W_{z}^{T}\right]=\left[\begin{array}{cc} R_{T} & 0\\ 0 & R_{P} \end{array}\right]\; \end{array}$$
Z(k+1) is the SD measurement vector, $W_{Z}\left(k+1\right)$ is the measurement noises, $R_{T}$ and $R_{P}$ represent, respectively, the convince matrix of the measurement noises $W_{T}$ and $W_{P}$. $R_{T}$ is non-diagonal matrixes due to DD operation on the measurements as shown in Equation (Equation 24). $R_{P}$ can be obtained similarly.

h(x(k+1)) is a nonlinear function that describes the relationship between the state vector and the measurement vector:(34)$$\begin{array}{c} h\left(x\left(k+1\right)\right)=\left[\begin{array}{c} h{\left(x\left(k+1\right)\right)}_{T}\\ h{\left(x\left(k+1\right)\right)}_{P} \end{array}\right]\;\\ h{\left(x\left(k+1\right)\right)}_{T}=\left[\begin{array}{c} h_{T}^{1j}\\ \vdots \\ h_{T}^{(j-1)j}\\ h_{T}^{(j+1)j}\\ \vdots \\ h_{T}^{mj} \end{array}\right],h{\left(x\left(k+1\right)\right)}_{P}=\left[\begin{array}{c} h_{P}^{1j}\\ \vdots \\ h_{P}^{(j-1)j}\\ h_{P}^{(j+1)j}\\ \vdots \\ h_{P}^{mj} \end{array}\right]\;\\ h_{T}^{ij}=h_{T}^{i}-h_{T}^{j};\left(i=1,\ldots ,m;i\neq j\right)\;\\ h_{P}^{ij}=h_{P}^{i}-h_{P}^{j};\left(i=1,\ldots ,m;i\neq j\right)\;\\ h_{T}^{i}=\sqrt{{\left(x\left(k+1|k\right)-x_{i}\right)}^{2}+{\left(y\left(k+1|k\right)-y_{i}\right)}^{2}};\left(i=1,\ldots ,m\right)\;\\ h_{P}^{i}=\frac{\sqrt{{\left(x\left(k+1|k\right)-x_{i}\right)}^{2}+{\left(y\left(k+1|k\right)-y_{i}\right)}^{2}}}{\lambda }-N_{ru}^{i};\left(i=1,\ldots ,m\right) \end{array}$$

There is a need to linearize the measurement Equation (Equation 30) around the estimated UE location to use the EKF algorithm. The Jacobian matrix H can be obtained as:(35)$$H\left(x\left(k+1|k\right)\right)=\frac{\partial h}{\partial x}{|}_{x\left(k+1|k\right)}=\left[\begin{array}{c} \frac{\partial h_{T}}{\partial x}{|}_{x\left(k+1|k\right)}\\ \frac{\partial h_{P}}{\partial x}{|}_{x\left(k+1|k\right)} \end{array}\right]$$
(36)$$\frac{\partial h_{T}}{\partial x}{|}_{x\left(k+1|k\right)}=\left[\begin{array}{cccc} \frac{\partial h_{T}^{1j}}{\partial x}{|}_{x\left(k+1|k\right)} & \frac{\partial h_{T}^{1j}}{\partial y}{|}_{y\left(k+1|k\right)} & \\ \vdots & \vdots & \\ \frac{\partial h_{T}^{(j-1)j}}{\partial x}{|}_{x\left(k+1|k\right)} & \frac{\partial h_{T}^{(j-1)j}}{\partial y}{|}_{y\left(k+1|k\right)} & 0(m-1\times 2) & 0(m-1\times m-1)\;\\ \frac{\partial h_{T}^{(j+1)j}}{\partial x}{|}_{x\left(k+1|k\right)} & \frac{\partial h_{T}^{(j+1)j}}{\partial y}{|}_{y\left(k+1|k\right)} & \\ \vdots & \vdots & \\ \frac{\partial h_{T}^{mj}}{\partial x}{|}_{x\left(k+1|k\right)} & \frac{\partial h_{T}^{mj}}{\partial y}{|}_{y\left(k+1|k\right)} \end{array}\right]$$
(37)$$\frac{\partial h_{P}}{\partial x}{|}_{x\left(k+1|k\right)}=\left[\begin{array}{cccc} \frac{\partial h_{P}^{1j}}{\partial x}{|}_{x\left(k+1|k\right)} & \frac{\partial h_{P}^{1j}}{\partial y}{|}_{y\left(k+1|k\right)} & \\ \vdots & \vdots & \\ \frac{\partial h_{P}^{(j-1)j}}{\partial x}{|}_{x\left(k+1|k\right)} & \frac{\partial h_{P}^{(j-1)j}}{\partial y}{|}_{y\left(k+1|k\right)} & 0(m-1\times 2) & -I(m-1\times m-1)\;\\ \frac{\partial h_{P}^{(j+1)j}}{\partial x}{|}_{x\left(k+1|k\right)} & \frac{\partial h_{P}^{(j+1)j}}{\partial y}{|}_{y\left(k+1|k\right)} & \\ \vdots & \vdots & \\ \frac{\partial h_{P}^{mj}}{\partial x}{|}_{x\left(k+1|k\right)} & \frac{\partial h_{P}^{mj}}{\partial y}{|}_{y\left(k+1|k\right)} \end{array}\right]$$
(38)$$\begin{array}{c} \frac{\partial h_{T}^{ij}}{\partial x}{|}_{x\left(k+1|k\right)}=\frac{\partial h_{T}^{i}}{\partial x}{{|}_{x\left(k+1|k\right)}-\frac{\partial h_{T}^{j}}{\partial x}|}_{x\left(k+1|k\right)};\left(i=1,\ldots ,m;i\neq j\right)\;\\ \frac{\partial h_{P}^{ij}}{\partial x}{|}_{x\left(k+1|k\right)}=\frac{\partial h_{P}^{i}}{\partial x}{{|}_{x\left(k+1|k\right)}-\frac{\partial h_{P}^{j}}{\partial x}|}_{x\left(k+1|k\right)};\left(i=1,\ldots ,m;i\neq j\right)\;\\ \frac{\partial h_{T}^{i}}{\partial x}{{|}_{x\left(k+1|k\right)}=\lambda \frac{\partial h_{P}^{i}}{\partial x}|}_{x\left(k+1|k\right)}=\frac{x\left(k+1|k\right)-x_{i}}{\sqrt{{\left(x\left(k+1|k\right)-x_{i}\right)}^{2}+{\left(y\left(k+1|k\right)-y_{i}\right)}^{2}}}\\ \frac{\partial h_{T}^{i}}{\partial y}{{|}_{y\left(k+1|k\right)}=\lambda \frac{\partial h_{P}^{i}}{\partial y}|}_{y\left(k+1|k\right)}=\frac{y\left(k+1|k\right)-y_{i}}{\sqrt{{\left(x\left(k+1|k\right)-x_{i}\right)}^{2}+{\left(y\left(k+1|k\right)-y_{i}\right)}^{2}}}\\ \frac{\partial h_{P}^{ij}}{\partial N_{ru}^{ij}}=-1 \end{array}$$

Given the state and measurement equations in previous sections, the EKF algorithm can be applied to calculate the estimate of x(k+1) based on the SD measurements. EKF algorithm [31,32] includes the following time-update and measurement update equations. Furthermore, the time-update equation are:(39)$$\begin{array}{c} x(k+1|k)=F(k)x(k|k)\;;\\ P\left(k+1|k\right)=F\left(k\right)P\left(k|k\right)F^{T}\left(k\right)+Q\left(k\right); \end{array}$$
where x(k|k) and P(k|k) are, respectively, the estimated state vector and its covariance matrix at the epoch $t=t_{k}$. x(k+1|k) and P(k+1|k) represent, respectively, the predicted state vector and its covariance matrix at the epoch $t=t_{k+1}$, based on x(k|k) and P(k|k). The matrixes F(k) and Q(k) are defined in Equation (Equation 29). Furthermore, the measurement update equation are:(40)$$\begin{array}{c} K(k+1)=P(k+1|k)H(x(k+1|k)){\left[H\left(x\left(k+1|k\right)\right)P\left(k+1|k\right)H^{T}\left(x\left(k+1|k\right)\right)+R\left(k\right)\right]}^{-1};\\ x(k+1|k+1)=x(k+1|k)+K(k+1)[Z(k+1)-h(x(k+1|k))];\\ P(k+1|k+1)=[I-K(k+1)H(x(k+1|k))]P(k+1|k); \end{array}$$

H(x(k+1|k)) is the Jacobian matrix given by Equation (Equation 35), the measurement equation h(x(k+1|k)) is defined in Equation (Equation 34), and calculated based on the predicted position (x(k+1|k),y(k+1|k)) at time $t=t_{k+1}$.

#### 3.2.1. NLOS Error Recognition and Elimination Based on EKF

Equation (Equation 25) shows that DD operation may not cancel out the impact of the NLOS. Furthermore, we propose an EKF-based scheme for NLOS error identification and elimination.
(41)$$\begin{array}{c} {\hat{T}}_{r}^{ij}≜cT_{ru}^{ij}+d_{u}^{ij}=d_{r}^{ij}+m_{ru}^{ij}+w_{ru,T}^{ij}\\ \Phi_{r}^{ij}≜\phi_{ru}^{ij}+\frac{d_{u}^{ij}}{\lambda }=\frac{d_{r}^{ij}+m_{ru}^{ij}}{\lambda }-N_{ru}^{ij}+w_{ru,P}^{ij} \end{array}.$$
According to the state and measurement equations at $t=t_{k}$, EKF can predict the SD measurements at $t=t_{k+1}$. Because the NLOS error reaches several meters, if there is NLOS propagation at $t=t_{k+1}$, the SD measurements will deviate greatly from the predicted value of EKF. NLOS error can be identified and corrected according to the deviation:(42)$$\begin{array}{c} \;\mathrm{if}\;\left|{\hat{T}}_{r}^{ij}-h_{T}^{ij}\right|>\Lambda \;\\ \;\mathrm{then}\;{\hat{T}}_{r}^{ij}=h_{T}^{ij},\Phi_{r}^{ij}=h_{P}^{ij} \end{array}.$$
The threshold setting depends on the maximum DD measurement noise. For the deviation greater than Λ, the NLOS error needs to be updated. The predicted measurements of EKF are used to improve the positioning accuracy.

#### 3.2.2. EKF Initialization

(43)$$x\left(0\right)={\left[x\left(0\right),y\left(0\right),v_{x}\left(0\right),v_{y}\left(0\right),N_{ru}^{1j}\left(0\right),\ldots ,N_{ru}^{mj}\left(0\right)\right]}^{T}.$$
For the first step of the EKF (t=0), the estimated initial UE position (x(0),y(0)) is obtained from the time difference of arrival (TDOA) or other approaches [33]. The initial estimates of $\left(v_{x}\left(0\right),v_{y}\left(0\right)\right)$ can be set to 0. The initial ambiguities $N_{ru}^{1j}\left(0\right),N_{ru}^{2j}\left(0\right),\ldots ,N_{r}^{mj}\left(0\right)$ can be simply determined based on the initial UE position and known positions of cell, i.e.,
(44)$$\begin{array}{c} N_{r}^{i}\left(0\right)=\frac{\sqrt{{\left(x\left(0\right)-x_{i}\right)}^{2}+{\left(y\left(0\right)-y_{i}\right)}^{2}}}{\lambda }-\phi_{r}^{i}\left(0\right);\\ N_{r}^{ij}=N_{r}^{i}-N_{r}^{j};\\ N_{ru}^{ij}=N_{r}^{ij}-N_{u}^{ij}; \end{array}$$
here, $N_{u}^{ij}$ is the SD integer ambiguity of the reference device. The initial covariance matrix $P_{0}$ can be set as the diagonal matrix as follows: (45)$$\begin{array}{c} \begin{array}{c} P\left(0\right)=\mathrm{diag}\left\{P_{x}\left(0\right),P_{y}\left(0\right),P_{v_{x}}\left(0\right),P_{v_{y}}\left(0\right),P_{N}\left(0\right)\right\};\\ P_{N}\left(0\right)=\left\{P_{N_{1j}}\left(0\right),\ldots ,P_{N_{(j-1)j}}\left(0\right),P_{N_{(j+1)j}}\left(0\right),\ldots ,P_{N_{mj}}\left(0\right)\right\}; \end{array} \end{array}$$
where $P_{x}\left(0\right)$, $P_{y}\left(0\right)$ can be set based on the assumed maximum positioning error of the TDOA. $P_{v_{x}}\left(0\right),P_{v_{y}}\left(0\right)$ can be set based on the expected maximum velocity of the UE; $P_{N_{1j}}\left(0\right),\ldots ,P_{N_{mj}}\left(0\right)$ are set based on the maximum assumed DD measurement error.

#### 3.2.3. Interaction with the Ambiguity Resolution Block

The EKF estimated float DD carrier-phase ambiguities would be sent to ambiguity resolution block to get integer DD carrier-phase ambiguities to improve positioning accuracy. For this purpose, after each EKF step *k*, the float solution of the DD carrier-phase ambiguities $\hat{N}\left(k|k\right)$ and the corresponding to covariance matrix $P_{N}\left(k\right)$ are provided to the ambiguity resolution block for searching the DD integer ambiguities $\bar{N}\left(k|k\right)$. To fix integer ambiguities, we use MLAMBDA, a modified LAMBDA method for integer least squares ambiguity determination [34,35].

DD integer ambiguities $\bar{N}\left(k|k\right)$ can be used to update $\hat{N}\left(k|k\right)$. However, the EKF performance may be degraded if unreliable $\bar{N}\left(k|k\right)$ is used to update $\hat{N}\left(k|k\right)$. Thus, before using $\bar{N}\left(k|k\right)$ to update $\hat{N}\left(k|k\right)$, there is a need to test the reliability of the DD integer ambiguities $\bar{N}\left(k|k\right)$.

The following approach is used to test the reliability of DD integer ambiguities $\bar{N}\left(k|k\right)$.

Initialization: Set a predefined threshold for the ratio test: ϵ>0, e.g., ϵ=0.5. Set a predefined maximum count $n_{max}$, e.g., $n_{max}=5$. Set counter n(0)=0.Step 1: For each epoch *k*, requesting the MLAMBDA to output two sets of the DD integer ambiguities. With the request, MLAMDA will return one group of the best estimates and one group of the second-best of the DD integer ambiguities and together with the corresponding residuals, say $r_{1}\left(k\right)$ and $r_{2}\left(k\right)$.Step 2: Calculate the ratio of $r_{1}\left(k\right)/r_{2}\left(k\right)$, and compared it with a predefined threshold ϵ. The smaller $r_{1}\left(k\right)/r_{2}\left(k\right)$ indicates that the best DD integer ambiguities estimates and the second-best estimates are close. If $r_{1}\left(k\right)/r_{2}\left(k\right)<\epsilon$, the counter n(k) is increased by 1, i.e., n(k)=n(k−1)+1. Otherwise, set n(k)=0.Step 3: If $n\left(k\right)>n_{max}$, declared that the $\bar{N}\left(k|k\right)$ is reliable DD integer ambiguity resolution. Once reliable DD integer ambiguity resolution $\bar{N}\left(k|k\right)$ is obtained, it can be used to update the EKF.

#### 3.2.4. Interaction with the Pre-Processing Measurement Block

Before each EKF operation, the EKF needs to adjust the state variables and covariance matrices based on TOA and carrier phase measurements.

If at time $t=t_{k}$, it is detected that there is a cycle slip for the phase measurements from *i*-th cell, the corresponding state, and covariance of the cell need to be reset. $N_{ru}^{ij}$ can be reset based on the TDOA measurement ${\hat{T}}_{r}^{ij}$ and the SD carrier phase measurement $\Phi_{r}^{ij}$. The diagonal element P(k|k) corresponding to $N_{ru}^{ij}$ will be set based on the maximum assumed integer ambiguities measurement error.

Suppose at time $t=t_{k}$, the measurements associated with an existing *i*-th cell are no longer available. In that case, the corresponding state $N_{ru}^{ij}$ needs to be removed from EKF, and so the elements of covariance matrix P(k|k) related the $N_{ru}^{ij}$. The dimension of the EKF will be reduced correspondingly.

If $t=t_{k}$, the measurements associated with a new cell are available, the EKF will add a new state of integer ambiguity for that cell. The corresponding state of the cell is calculated based on the TDOA measurement ${\hat{T}}_{r}^{ij}$ and the SD carrier phase measurement $\Phi_{r}^{ij}$. The diagonal element of P(k|k) corresponding to $N_{ru}^{ij}$ will be set based on the maximum assumed integer ambiguities measurement error.

Figure 2 shows the signal processing diagram for the real-time kinematic positioning based on TOA and carrier phase measurements.

## 4. Numerical Results

In this paper, MATLAB is used to verify the algorithm. Furthermore, one PRS subframe is used in each PRS positioning occasion. Perfect muting is assumed in the simulation. The positioning scene is shown in Figure 3, where six BSs are regularly distributed in the building, and the reference device is located in the center of the scene. In the simulation, it is assumed that there is a synchronization error in Network. Therefore, there are time-varying synchronization errors at the BS side and the terminal side. The detailed simulation parameters are listed in Table 1. For other parameters including the number of multipath in the indoor scenarios, the criteria for generating LOS/NLOS, and the path loss, please refer to [36].

In the simulation, moving speed of the terminal is 1 m/s, and the position solution interval is 0.1 s. The terminal moves according to a specific track in which the length is 60 m, and the number of sampling epoch is 600. High accuracy localization using the carrier phase requires a fast and accurate solution of the integer ambiguity. Therefore, we use the following perspectives to evaluate the effectiveness of the algorithm:The accuracy and convergence speed of the integer ambiguity.The terminal that can judge whether the solved integer ambiguity is reliable or not.The real-time positioning accuracy of the terminal position.The cumulative distribution curve of positioning error.

Equation (Equation 25) shows that the SD carrier phase measurement $\Phi_{r}^{ij}$ contains the DD integer ambiguity $N_{ru}^{ij}$. Therefore, we use the DD integer ambiguity for performance comparison. Define the integer ambiguity estimation error as:(46)$$e_{N}=\left|N_{ru,ture}^{ij}-N_{ru}^{ij}\right|$$

Figure 4 illustrates the four DD integer ambiguity estimation errors. When $e_{N}=0$, it represents that the estimated integer ambiguity is the same as the actual ambiguity. Furthermore, the first BS is used as the reference BS in our experiment. All integer ambiguity errors were significant at epoch 0 due to the sizeable initial position estimation error. In the 92nd epoch, BS21, BS41, and BS51 all estimate the integer ambiguity correctly and remain unchanged in the subsequent epochs; BS31 always has an error of 1 circle during the experiment. The mistake of BS31 did not affect the localization accuracy because the other integer ambiguities were correctly estimated.

Figure 5 shows the ratio test was used to check whether the DD integer ambiguities output by the EKF are reliable at the current epoch. The dashed line represents the preset threshold ϵ=0.5. After the 113th epoch, the reliability rates are all below the threshold. Therefore the algorithm determines that the obtained integer ambiguities are reliable after the 113th epoch. Combined with Figure 4, it can be seen that it is valid to use the ratio test to determine whether the DD integer ambiguities converge to the actual value.

We evaluate the performance of the `TOA+CP EKF’ based differential positioning method as in Figure 6. In addition, we also list two other cases for comparison, wherein `GMM EKF’ [31] is a method to perform positioning solution by TDOA measurement, which eliminates the NLOS error by model NLOS propagation as Gaussian mixture model; `EKF’ [32] represents a commonly used EKF location algorithm based on TDOA measurement. To eliminate the effect of clock errors, all three algorithms mentioned above use TDOA obtained from Equation (Equation 25) instead of TOA for positioning, and the algorithm proposed in this paper also requires SD carrier phase measurements.

Figure 6 and Figure 7 show the performance of the three algorithms during mobile localization. Since all three algorithms use differential measurements for position solving, it can be seen from the results that the time-varying synchronization errors do not affect the positioning accuracy. At the initial epoch, the accurate integer ambiguity has not been solved, so the carrier phase measurement is difficult to determine the initial position. Therefore, initial positions of all three algorithms are calculated from the Chan algorithm [33] using TDOA measurements. It can be seen from Figure 6 and Figure 7 that in the first few epochs, the positioning error of `TOA+CP EKF’ is significant, which is caused by the inaccurate integer ambiguity. In subsequent periods, as the algorithm correctly fix the integer ambiguity, the positioning error gradually decreases. Furthermore, the carrier phase measurement is not limited by the system sampling rate, which, combined with the correct integer ambiguity, makes the carrier phase algorithm suitable for scenarios with high accuracy requirements. Comparatively, both `GMM EKF’ and `EKF’ use only TDOA for user position tracking, which leads to lower positioning accuracy.

The cumulative density function (CDF) curves of horizontal positioning errors are used as performance metrics in positioning evaluations. Define the positioning error as:(47)$$e_{pos}=\sqrt{{\left(\hat{x}-x_{\mathrm{ture}}\right)}^{2}+{\left(\hat{y}-y_{\mathrm{ture}}\right)}^{2}}$$

The CDFs for the localization error from both methods are shown in Figure 8. The `TOA+CP EKF’ method has the best performance, with 90% of the horizontal positioning errors within 0.27 m. Therefore, the carrier phase-based localization technique can meet the high accuracy localization requirements. The `GMM EKF’ method has the middle performance due to the algorithm using TDOA for user location tracking and NLOS elimination. Since the system sampling rate limits the TDOA measurement resolution, the positioning accuracy is low. `EKF’ method has the worst performance because it only uses the TDOA and has limited effectiveness in eliminating NLOS error.

We simulated the localization accuracy of this algorithm with the different number of BSs. In our experiments, as shown in Figure 9, we set the length of the indoor scenario to 100 m and the width to 20 m. Furthermore, the coordinates of the six BSs are [0,0],[40,0],[100,0],[0,20],[40,20],[100,20], respectively. The coordinate of the reference UE is [50,10]. The actual distance between the user and the BS determines the probability of LOS. Thus, the expansion of the simulation environment decreases the LOS probability and equivalently simulates the case of increasing obstacles.

When five BSs are used in the experiment, the BS located at [40,0] is removed. When four BSs are involved in localization, the two BSs located at [40,0],[40,20] are removed. From Figure 10, it can be seen that the localization accuracy of the algorithm proposed in this paper decreases as the number of BSs decreases. The decrease in the number of available BSs leads to a more extended solution period for the integer ambiguity and thus decreases the localization accuracy. In addition, compared with Figure 8, the decrease in LOS probability does not cause severe degradation of the localization accuracy, so the NLOS error suppression scheme proposed in this paper is effective.

## 5. Conclusions

The main research direction of this paper is to apply carrier phase technology in OFDM systems to improve ranging and positioning accuracy. Compared with single-point positioning using only TOA measurement, carrier phase information is more accurate than TOA measurement, and it is a possible choice for indoor high-precision positioning. This paper intends to solve two problems of indoor carrier phase positioning: 1. Phase measurements in a multipath environment. 2. Fast and precise integer ambiguity resolution in real-time positioning scenarios. First, this paper analyzed the effect of multipath propagation on phase measurement in detail, and proposed a correlation profile-based carrier phase measurement method. Second, this paper presents an EKF algorithm to estimate the integer ambiguity by the SD carrier and TDOA measurements. In addition to the integer ambiguity estimation, the algorithm also considers the effect brought by NLOS error. Experiments show that the algorithm proposed in this paper can quickly find the integer ambiguity and virtually eliminate the NLOS error, thus improving the positioning accuracy.

## Figures and Tables

**Figure 1 sensors-21-06731-f001:**
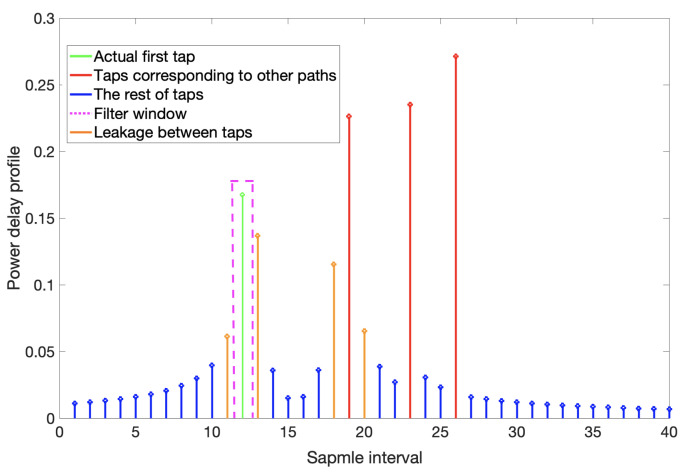
Power delay profile.

**Figure 2 sensors-21-06731-f002:**
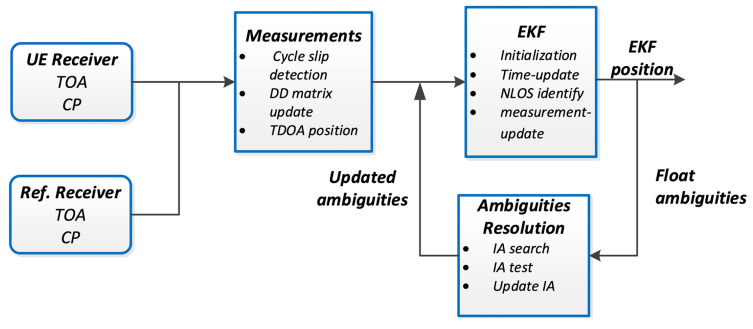
Flowchart of TOA/Carrier Phase combined Real-Time Kinematic.

**Figure 3 sensors-21-06731-f003:**
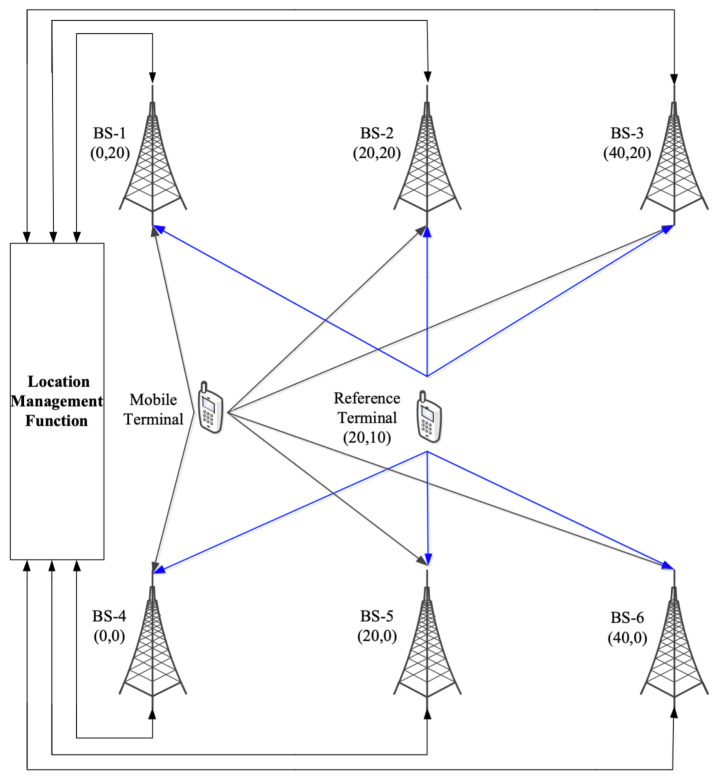
Terminal positioning environment.

**Figure 4 sensors-21-06731-f004:**
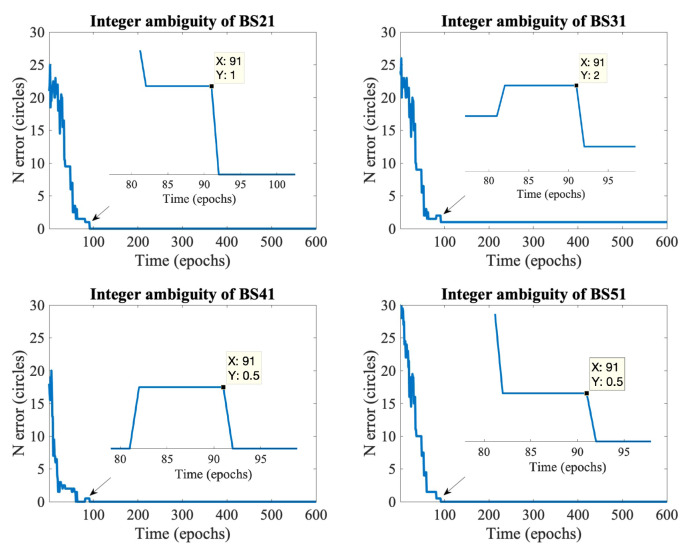
Schematic diagram of DD integer ambiguity convergence.

**Figure 5 sensors-21-06731-f005:**
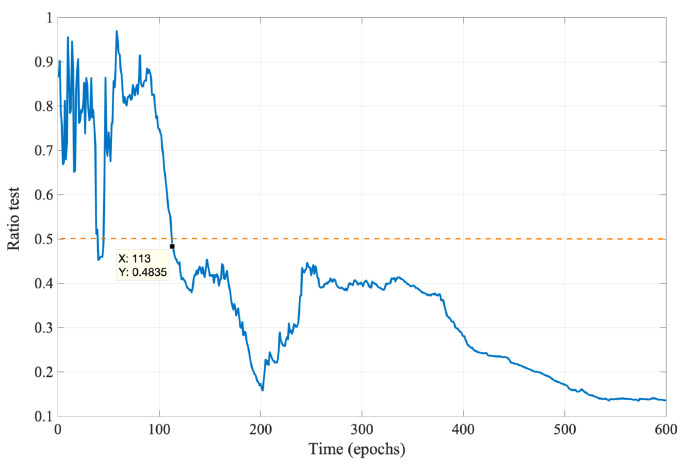
Test of the reliability of DD integer ambiguities.

**Figure 6 sensors-21-06731-f006:**
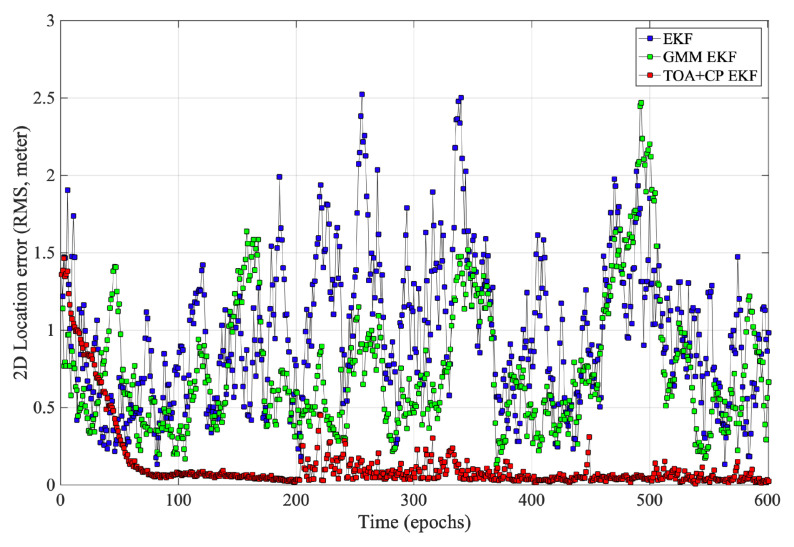
Statistic of mobile positioning error.

**Figure 7 sensors-21-06731-f007:**
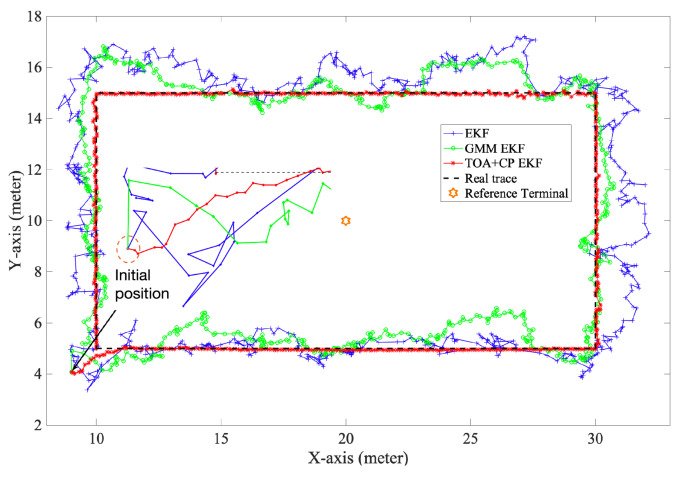
Localization performance of EKF.

**Figure 8 sensors-21-06731-f008:**
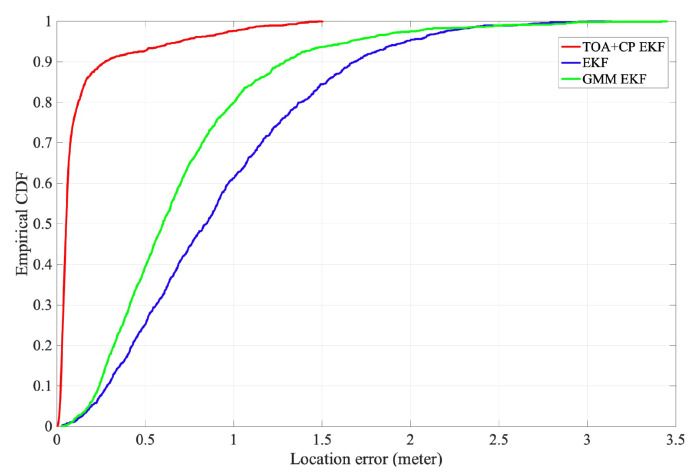
The CDF of horizontal localization error.

**Figure 9 sensors-21-06731-f009:**
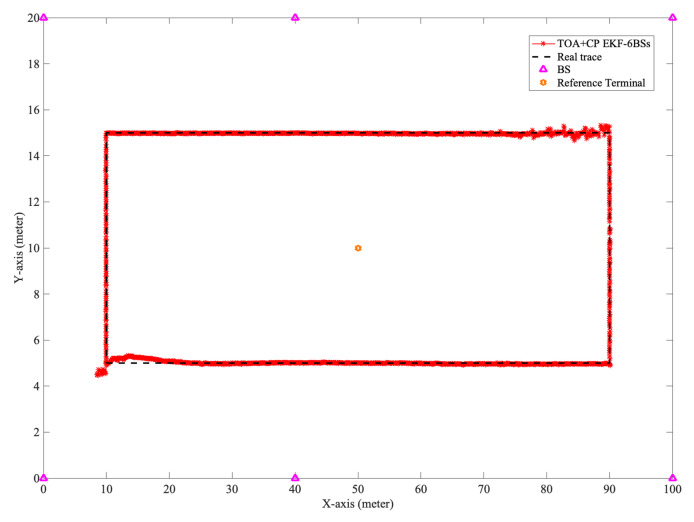
Layout of Indoor - Mixed office scenario.

**Figure 10 sensors-21-06731-f010:**
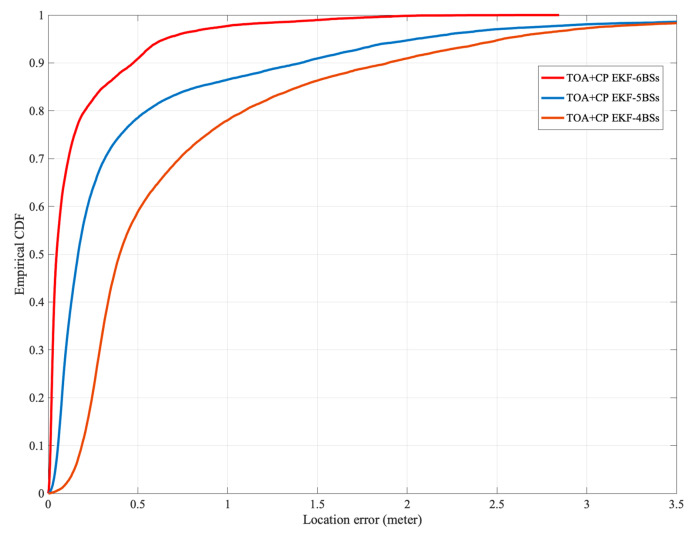
The CDF of horizontal localization error with differfent number of BSs.

**Table 1 sensors-21-06731-t001:** System Parameters.

Parameters	Values
Channel model	5G New Radio (NR) channel model (Indoor-Mixed office [36]).
Carrier frequency	3.5 GHz
Carrier wavelength	0.085 m
Inter-site distance	20 m
Room size	40 m × 20 m
Subcarrier spacing	15 KHz
Reference signal	New Radio PRS Structure from [37].
Reference Signal Transmission Bandwidth	50 MHz
Number of BSs	6
UE-antennas	4
Number of subcarriers	3240
FFT Length	4096 for 50 MHz
Sampling rate	61.44 MHz for 50 MHz
Number of occasions used per positioning estimate	1
Interference modelling	Perfect muting
Clock error between BSs	Gaussian distribution with a mean of 25 ns and a variance of 10 ns.
Clock error of the terminal	Gaussian distribution with a mean of 50 ns and a variance of 15 ns.
Delay spread	Exponential distribution with a mean of 22 ns.
Total transmission power	24 dBm
Maximum directional gain of an antenna element	5 dBi
UE speed	1 m/s
Position solution interval	0.1 s
NLOS error identification threshold	Λ=3
Ratio test threshold	ϵ=0.5
LOS generation probability	Table 7.4.2-1 in the literature [36].
Fading model	Large scale fading: Table 7.4.1-1 in the literature [36]; Fast fading: Section 7.5 of [36].
Channel independence	The channel model of the reference device and the channel model of the user terminal are independent of each other.

## Data Availability

The data presented in this study are available on request from the corresponding author. The data are not publicly available due to privacy reasons.

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
