# Peer review of "Indoor Carrier Phase Positioning Technology Based on OFDM System"

_sensors, 2021, doi:10.3390/s21206731_

Round 1

Reviewer 1 Report

The authors propose a ranging scheme based exploiting carrier phase measurements of a OFDM system, focusing on multipath and NLoS propagation and integer ambiguity resolution.

(1) In the abstract, the authors state that " carrier phase ranging [...] is not used for wireless orthogonal frequency division multiplex (OFDM) techniques".
However, various other research papers deal with using the carrier phase of OFDM/LTE signals for localization, for example references [16-18] or:
K. Shamaei and Z. Kassas, "Sub-meter accurate UAV navigation and cycle slip detection with LTE carrier phase measurements" ION Global Navigation Satellite Systems Conference, Sep. 16-20, 2019

Please clarify and state the fundamental differences between your paper and these papers.

(2) The authors assume that a reference terminal with known location is available. This reference terminal is pivotal for the entire ranging method to work. This reference terminal is a strong assumption on the ranging system, since additional hardware needs to be installed, and the position of this reference terminal needs to be known exactly. The practical relevance of the authors' method is therefore very limited.

Is there any way to weaken this assumption? For example by  
o assuming that one of the base stations serves as reference terminal?
o jointly estimating the position of the user terminal, the reference terminal and the clock error? In that case, it would be a cooperative method.
If so, what are the impacts on the system performance?

In my opinion, this is the critical point in the paper, and it should be addressed with care.

(3) Diffuse multipath components (DMC)* are neither considered in the theoretical derivation (Equations (3)/(4)) nor in the practical evaluation of the paper. However, in my understanding, DMC are a critical issue in relatively narrow indoor environments.

Please clarify.
Are there DMC considered in reference [30]?

* For more information on DMC, see for example:
A. Richter, "Estimation of radio channel parameters: models and algorithms", 2005, PhD Thesis, Ilmenau University of Technology

(4) The paper lacks an evaluation with real measurement data. Thus, knowing the simulation environment and parameters is very important to evaluate the performance of the ranging/localization system. However, the environment and parameters are not clear to me. In particular, the authors refer to reference [30] for further simulation parameters. In this reference, multiple choices for channel model parameters are described.

Referring to reference [30], please give more information on your channel model, for example:
o do you assume an "Indoor - Mixed office" or "Indoor - Open office" model?
o which penetration loss / which wall material is assumed?
o are fading and blockage incorporated? if so, how?

Please also clearly state the probabilities of LoS and NLoS for each base station in the paper, for example in Table 1. These parameters are of crucial importance for the interpretation of the simulation data.

Is the channel model assumed for the reference terminal the same as for the user terminal? 
Are the two channels for the two terminals to each BS assumed independent from each other? For example, if there is a NLoS condition from BS1 to the user terminal at some point, does that mean that there is a NLoS condition to the reference terminal as well?

(5) Further remarks

  • Please proofread the manuscript with regard to grammar and typos.
  • In the introduction, the authors state that they "utilize the EKF to eliminate NLOS propagation".
    NLOS propagation cannot be eliminated, only its effects. Please rephrase.
  • In Section 2.2, the authors state that "The phase shift [...] consists of a component due to free space propagation". This sentence is unclear, please clarify.
  • In Section 2.2, the authors state that "[...] the phase after the time-domain window has received some interference". This sentence is unclear, please clarify.
  • In Equation (7), it is not clear what exactly R_xy is. It is an expression, not a detector. Please be precise in wording.
  • In Equation (8), w is not defined.
  • In Equation (10) and in the first line of Equation (20), T_r^i is a ToA measurement, but the unit is meter. Please be exact in the wording.
  • Please work on the presentation of Figure 4. It is hard to interprete. For example, instead of vertical lines and markers, just use a line connecting the markers.
  • In Figure 4, the number of epochs for BS41 is different from the other base stations. Please clarify.
  • The threshold in Equation (42) is set to 0.5 in the evaluations (e.g., Figure 5). The authors state that this value "depends on the maximum DD measurement noise". Nevertheless, the value of 0.5 seems arbitrary. Is there any well-founded justification for this choice, for example from the measured/simulated data? 

Reviewer 2 Report

The proposed algorithm to estimate the integer ambiguity in carrier phase positioning is an interesting extension to the state of the art and clearly shows a new approach in IPS based on OFDM system. It is well described in the manuscript.

The paper is very well written, there are no major language errors. I was not able to identify any errors in the description of the proposed algorithm and in the derivations of the formulas. There are only some minor editing errors - e.g. there are two capital letters in the sentence starting with "And When" on line 113, p. 5. Additionally, some technical terms are used without appropriate explanation - eg. the paper would be more clear if a description of taps would be added to the text on lines 119-125 on p. 5.

The abstract is correctly defining the scope of the paper, however, it is lacking some information on how the proposed method improves the localization accuracy and what are the results of the evaluation of the proposed algorithm.

The review of the state of the art is appropriate, although it is quite broad, describing even the papers related to AoA and RSS techniques.

The model used for the validation of the algorithm is well built upon the 5G channel model and the selection of parameters is appropriate. However, it is based on a very optimistic scenario, with a large number of BSes (6) located in a very small area. Thus, although it is presenting a very significant accuracy increase, it is not clear how those results may be applicable in more realistic scenarios, with obstacles or a lower number of reference points. It would be recommended to extend the paper with an evaluation of the algorithm in few simulation scenarios e.g. with different numbers of reference points, showing how the accuracy varies.

Round 2

Reviewer 1 Report

I think that the authors have improved the manuscript considerably since the first submission. There is just one issue that I have left:

The authors do not really seem to have properly addressed my issue (3.) raised regarding dense multipath components (DMC).
They state that they have "have re-edited the equations involving DMC, as detailed in equations (3)-(7), (11)-(15), (17), (18)."
The only difference that I can see in these equations compared to the original manuscript is that a time dependency of the CIR.
However, no DMC are incorporated. DMC may cause the noise term (e.g. w^m in Eq.(4)) to be colored, for example.
Throughout the revised paper, I do not see any notion of DMC, or anything like that. 
Please clarify. 
